# Micropropagation as a Tool for the Conservation of Autochthonous *Sorbus* Species of Czechia

**DOI:** 10.3390/plants12030488

**Published:** 2023-01-20

**Authors:** Jana Šedivá, Jiří Velebil, Daniel Zahradník

**Affiliations:** Silva Tarouca Research Institute for Landscape and Ornamental Gardening, Public Research Institute, Květnové náměstí 391, 252 43 Průhonice, Czech Republic

**Keywords:** endemic species, micropropagation, ex vitro rooting, substrate, *Sorbus*

## Abstract

Members of the genus *Sorbus* are the only endemic tree species that occur in Czechia. They are important components of endangered plant communities. Their natural regeneration is usually problematic because of their mode of reproduction and because they can survive in rare populations with small numbers of individuals. The aim of this study was to develop a successful micropropagation protocol for selected *Sorbus* species, of which two are endemic (*S. gemella* and *S. omissa*) and two are hybrid (*S*. × *abscondita* and *S*. × *kitaibeliana*). We found significant differences in shoot induction and rooting ability between the *Sorbus* species under study. With the exception of *S.* × *abscondita*, *N*^6^-benzyladenine had a significantly greater effect on shoot regeneration, both in terms of shoot number and total shoot length, than *meta*-topolin. Root induction was key to the successful micropropagation of the *Sorbus* species studied. Our results show that four *Sorbus* species can be successfully rooted under ex vitro conditions, without a rooting powder treatment in a steamed peat-perlite substrate. Auxin-untreated microcuttings of *S. gemella*, *S.* × *kitaibeliana* and *S. omissa*, but not *S. × abscondita,* rooted better than ones treated with indole-3-butyric acid. This is the first time a micropropagation protocol for *S*. *omissa*, *S. × abscondita* and *S.* × *kitaibeliana* has been published.

## 1. Introduction

In the flora of Europe, the genus *Sorbus* L. is one of the most taxonomically challenging groups of woody plants. Taxonomical complications stem chiefly from the vast morphological variability and huge genetic diversity of the genus, which is generated by frequent interspecific hybridization, genome duplication and subsequent stabilization of offspring by apomixis [1,2,3,4,5]. Furthermore, diverse opinions on taxonomic treatment [6,7] and a tangled nomenclature still cause disagreements and misinterpretations.

Most *Sorbus* species are polyploids that are locally distributed in geographically clearly defined areas, and some are stenoendemics occurring in a single locality with one population or a few subpopulations. All polyploid apomictic *Sorbus* taxa in Central Europe are derived from hybrids of members of *Sorbus* subgen. *Aria* Pers. (including diploid *S. aria* (L.) Crantz and *S. umbellata* (Desf.) Fritsch and polyploid apomictic taxa) or from hybrids between *S.* subgen. *Aria* and members of one of the three following subgenera: *S.* subgen. *Sorbus* (diploid *S. aucuparia* L.), *S.* subgen. *Torminaria* (DC.) Reichenb. (diploid *S. torminalis* (L.) Crantz) and *S.* subgen. *Chamaemespilus* (Medik.) K. Koch (diploid *S. chamaemespilus* (L.) Crantz). Furthermore, the same parental combination can polytopically form various apomictic, morphologically homogeneous and genetically unique lines which are classified in modern taxonomy as separate species (i.e., microspecies or agamospecies). As well as the species mentioned above, another diploid, *S*. *domestica* L. (*S.* subgen. *Cormus* (Spach) Duch.), occurs in Europe. Diploids are sexual and are rather widely distributed. In rare instances, they form diploid, sexual and variable offspring (*S. aria* × *S. aucuparia* = *S.* × *thuringiaca* (Nyman) Fritsch, *S. aria* × *S. torminalis* = *S.* × *decipiens* (Bechst.) Petz. & Kirchn., *S*. × *ambigua* (Decne.) Beck = *S. aria* × *S. chamaemespilus*).

According to current knowledge, there are 189 *Sorbus* species in Europe [8], but taxonomical research using contemporary biosystematic methods is still revealing novelties in this field [5,9,10,11,12], and the number of species keeps increasing. Most of them are threatened endemics listed in the European Red List of Trees [13]. In Czechia, there are 21 native *Sorbus* species, of which 13 are endemics and three are subendemics (*S. cucullifera* M. Lepší & P. Lepší; *S. sudetica* (Tausch) Bluff, Nees & Schauer; *S. thayensis* M. Lepší & P. Lepší) ([14,15]; Figure 5). Populations of (sub)endemics are not very numerous. They usually consist of only a few dozen (in extreme cases, less than 20) or, at most, hundreds of individuals. Most of them are located on sites representing residual fragments of natural biotopes in the cultural landscape, the protection of which is insufficient or even lacking altogether. In the last version of the Red List of vascular plants of the Czech Republic, they are included in the ‘threatened’ category [16].

The natural regeneration of *Sorbus* species limits their survival and further prosperity in natural habitats. In nature, they reproduce mainly generatively. To a limited extent, they can also spread clonally via rooted branches on avalanche paths or scree slopes (e.g., *S. sudetica*, and *S*. × *decipiens*; [17,18,19]) or via root offsets (*S.* × *decipiens*; own observation). They are entomogamous and mostly allogamous, so the successful development of their seeds depends on the presence of a greater number of plants within the flight range of insects. Autogamy (i.e., self-pollination) occurs only rarely (recorded in *S. aucuparia* and *S. torminalis*; [2,20,21]). As already stated, most of the species are polyploids with an apomictic mode of reproduction. All polyploid species are capable of pseudogamic reproduction [22,23,24,25]. It is therefore necessary to ensure the presence of other species (pollen donors) in order for viable seeds to form under natural conditions.

All Czech species of *Sorbus* are light-demanding and prefer open habitats such as rocks and screes, rock steppes, scrubs, woodland-steppes and thermophilous open pine, oak or hornbeam and ravine woodlands, as well as their fringes. As well as being semi-natural to relic vegetation, they occasionally grow in *Larix decidua*, *Picea abies*, *Pinus nigra*, *P. sylvestris* and, rarely, *Robinia pseudoacacia* plantations or in their clearings. The closed canopy that now prevails in woodlands in Czechia is unfavorable for the long-term survival and regular reproduction of such species. They are also potentially threatened by the disappearance of habitats, as open woodlands develop a dense canopy or are replaced by forest plantations [14].

The conventional method using seeds is most commonly used to reproduce *Sorbus* species [26]. Vegetative propagation by grafting or budding is used less frequently. Furthermore, this method is not suitable for endangered *Sorbus* species. The use of cuttings is inefficient compared to micropropagation. Plant tissue cultivation can provide a rapid method for the multiplication of endangered or threatened species when only a few stock plants are available or if the collection of plants and seeds from wild plants needs to be minimized [27].

A micropropagation protocol was first developed for economically important *Sorbus* species such as *S. torminalis* [28,29,30], *S. aucuparia* [29,31,32,33] and *S. domestica* [28,34,35,36,37,38,39,40]. These species are mainly used for wood production, as well as in the food industry and in ornamental horticulture. Research carried out in the last decade has shown that micropropagation is an important method for the propagation of rare and endemic *Sorbus* species or their infrageneric hybrid such as ×*Malosorbus florentina* [41], and then *S*. *alnifrons*, *S*. *bohemica*, *S*. *gemella* Kovanda, *S. hardeggensis*, *S. quernea*, *S*. *rhodanthera* and *S*. *sudetica* [42]. These species are important primarily from an ecological point of view, as they increase the biodiversity of woodland ecosystems. Some endemic species of *Sorbus* are found and grow well even in extreme conditions (with some degree of drought resistance), where common tree species do not grow. Some *Sorbus* species have been discovered only recently or little is known about them. In this study, we focused on the establishment of a micropropagation method of *S. gemella*, *S. omissa* Velebil, *S*. × *abscondita* Kovanda and *S*. × *kitaibeliana* Baksay & Kárpáti. The characteristics of *Sorbus* species used in this study and their localization are presented in Section 4, i.e., Materials and Methods.

## 2. Results

### 2.1. In Vitro Shoot Proliferation

The occurrence of explant contamination was low, ranging from 0 to 15%. Satisfactory reactivity of the explants to the in vitro cultures was achieved and reached a total of 79%. In our experiments, initial cultures of *Sorbus* species were established from axillary buds originating from the lower branches of donor trees. Better regeneration of shoots and roots could be expected in the subsequent stages of in vitro culture. This effect was observed in *S. aucuparia*, when microshoots derived from the lower branches of mature trees exhibited better shoot proliferation and rooting response than from the top branches [43]. After 6 months of cultivation, in vitro cultures of *Sorbus* species were stabilized.

There were significant differences (*p*-value < 0.05) between the *Sorbus* species under study in terms of the mean number of shoots per explant (Figure 1A). *Sorbus × abscondita* (2.71) and *S*. *× kitaibeliana* (2.35) produced significantly more shoots than *S. gemella* (1.88) and *S. omissa* (1.86).

The application of cytokinins during multiplication significantly increased shoot regeneration compared to the control (1.03; Figure 1C). A higher shoot production (4.17) was observed on the multiplication medium with *N*^6^-benzyladenine (BA) than the one with *meta*-topolin (*m*T, 1.58). The combination of each of the cytokinins with the auxin indole-3-butyric acid (IBA) had no statistical effect on the mean number of shoots.

Another growth parameter was total shoot length, in which significant differences were found between *Sorbus* species (Figure 1B); the response was similar to that for the mean number of shoots per explant (Figure 1A). *Sorbus × abscondita* (61.48 mm) and *S*. *× kitaibeliana* (53.72 mm) produced significantly longer shoots than *S. omissa* (38.47 mm) and *S. gemella* (36.07 mm).

The cytokinins also had a positive effect on total shoot length compared to the control (20.67 mm; Figure 1D). The shoots on the multiplication medium with BA were significantly longer (77.89 mm) than that with *m*T (39.61 mm). The combination of each of the cytokinins with the auxin (IBA) had no statistical effect on total shoot length.

The interaction between the genotype and the PGR treatment showed a different response in shoot production in three *Sorbus* species (Table 1). In *S. × abscondita*, the type of cytokinin played a significant role, and shoot formation was significantly higher on the medium with *m*T (3.72) or *m*T + IBA (3.47) than on that with BA (2.97). In *S. gemella*, shoot formation was significantly higher on the medium with *m*T + BA (2.69) compared to the control (1.11) and the IBA treatment (1.03). In *S. × kitaibeliana*, shoot formation was significantly higher on the medium with BA (3.42) compared to the control (1.28) and IBA treatment (1.36).

Regarding the interaction between the genotype and the PGR treatment on total shoot length, significant differences were observed in two *Sorbus* species (Table 2). In *S. × abscondita*, shoot elongation was significantly better in the *m*T + IBA treatment (89.22 mm) than in the combined BA + IBA treatment (54.94 mm) and the IBA treatment (39.99 mm). In *S*. *× kitaibeliana*, the combination of *m*T + IBA (65.50 mm) was also effective, as was the BA treatment (71.00 mm), where the total length of the shoots was significantly higher compared to the control (31.69 mm) and the IBA treatment (36.31 mm).

### 2.2. Ex Vitro Rooting and Acclimatization

The effects of different factors (genotype, auxin treatment/commercial rooting powder and microcutting size) on rooting were examined. All observed factors had significant effects on the rooting percentage and the mean number of roots per microcutting at different significance levels. The effects of interactions between two factors were found to be significant for genotype × auxin treatment for all growth characteristics and genotype × microcutting size for rooting percentage (Table 3).

Rhizogenesis was observed in all *Sorbus* species. There were statistically significant differences in rooting between *Sorbus × abscondita, S. gemella, S. × kitaibeliana* and *S. omissa* (Figure 2A and Figure 3). A high rooting percentage was found in *S*. *× abscondita* (74%) and *S. omissa* (62%), whereas a low root capability was observed in *S. × kitaibeliana* (27%) and *S. gemella* (14%). A similar trend was also observed for the mean number of roots per microcutting (5.78, 3.18, 1.98 and 2.51, respectively; Figure 2B).

Root production also depended on the presence and the concentration of IBA in the rooting powder (Figure 2C). Auxin treatments (1% and 2% IBA) reduced the rooting percentage significantly (40% and 32%) compared to the control (61%). On the other hand, the auxin significantly increased the mean number of roots per microcutting, both in the 1% IBA treatment (4.66) and in the 2% IBA treatment (5.12.) compared with the control (2.91), see Figure 2D.

The size of the microcutting was also an important factor affecting *Sorbus* rooting (Figure 2E). We found that microcuttings that were 15–25 mm long rooted significantly better (54%) than microcuttings that were 26–35 mm long (34%). The opposite effect was observed for the mean number of roots per microcutting; in longer microcuttings, the formation of roots was significantly higher (4.66) than in shorter microcuttings (3.56), see Figure 2F.

The interaction between the genotype and the auxin treatment showed a different response in the rooting percentage of *Sorbus* × *abscondita* compared to other *Sorbus* species (Figure 2G). Only in this species was there a significant difference in rooting percentage between the control (81%) and the 2% IBA treatment (90%). The effect of auxin on the mean number of roots per microcutting was demonstrated conclusively in three *Sorbus* genotypes (Figure 2H). The auxin treatment increased the mean number of roots in *S*. × *abscondita* (8.0 and 6.59 with 1% and 2% IBA, respectively) and in *S. omissa* (1% IBA, 3.32) and had an antagonistic effect in *S. gemella* (2% IBA, 0.0) compared to the control. The effect of the interaction between the genotype and the size of the microcuttings on rooting percentage was significant only in *S. omissa*, where shorter microcuttings rooted better (84%) compared to longer microcuttings (40%; Figure 2I).

The plant acclimation process started during ex vitro rooting and was successfully completed after transplanting the plants to a greenhouse (Figure 3 and Figure 4).

## 3. Discussion

In preservation programs that specialize in the propagation of endangered tree species, often only adult trees are available as a source plant. The success of micropropagation is strongly influenced by the ontogenetic and chronological age of source plants. According to [27], ontogenetic aging (physiological) refers to the phases of development that the seedling plant undergoes from embryonic to juvenile to intermediate to mature (adult), and chronological aging continues through the life of an individual plant whether as a seedling or vegetatively propagated. In vitro cultures initiated from source plants that are physiologically mature or less vital (old) or both may exhibit growth depression during micropropagation.

In vitro cultures can be successfully obtained from an adult tree in several ways: establishing cultures from plant parts showing juvenile characteristics, from mature parts after rejuvenation or from serial subcultures of explants in a hormone medium [44,45,46].

Plant growth regulators play a key role in adventitious shoot regeneration in woody species of the temperate zone [47]. The presence of a cytokinin is necessary for de novo shoot formation in woody species, and very often a cytokinin is combined with auxin at a low concentration to increase the number of shoots.

BA is an important aromatic cytokinin that is very effective at shoot induction in many plant species, including *Sorbus* species, and it is affordable compared to other cytokinins. On the other hand, it can cause growth abnormalities in some plant species, reduce rooting and worsen the subsequent acclimatization of plants [30,48,49]. For these reasons, new aromatic cytokinin derivatives have been the subject of intensive research [50]. One of them is *m*T, a natural highly active aromatic cytokinin occurring in poplar leaves [51]. *Meta*-topolin has been successfully used in the micropropagation of many plant species, including both herbs [49,52,53] and trees [54,55,56]. Ördögh et al. [57] published the effects of different types of cytokinin, including *m*T, on shoot proliferation in *S. borbasii* Jáv.

In our study, we compared the effectiveness of two cytokinins, BA and *m*T, singly or in combination, with auxin in four *Sorbus* species. In agreement with the results of Nikolaou et al. [37] and Jeong and Sivanesan [58], the presence of a cytokinin was essential for shoot proliferation in *Sorbus* species. In general, BA had a significantly higher effect on shoot regeneration, both in terms of shoot number and total shoot length, than *m*T (Figure 1C,D). This finding that BA increases the number of shoots more effectively than *m*T is consistent with the results obtained by Ördögh et al. [59] for *S. redliana* Kárpáti, by Malá et al. [30] for *S. torminalis,* by Ördögh et al. [57] for *Sorbus borbasii* and by Meyer et al. [60] for *Hypericum* L. species. On the other hand, some *Sorbus* species behaved differently: In *S.* × *abscondita*, shoot formation was significantly more vigorous on the multiplication medium with *m*T than the one with BA (Table 1 and Table 2). A similar stronger reaction to *m*T in shoot proliferation was observed in *Ribes* [61]. Likewise, Hlophe et al. [62] noticed that each *Brachystelma* R. Br. species differed in its response to specific cytokinins.

The combination of each of the cytokinins with the auxin IBA or the auxin alone had no significant effect on growth parameters in the *Sorbus* species studied (Figure 1), although a slight improvement in shoot production was observed in *S. gemella* and *S. omissa* on the medium with *m*T + IBA (Table 2). Similar results were obtained with *S. domestica*, where the addition of IBA to BA had no effect on shoot proliferation [37]. Many authors routinely used a combination of a cytokinin and the auxin IBA for proliferation without further analyses [30,41,43,57,59]. In contrast to our results, it has been reported that the combination of a cytokinin (BA) and auxin (IAA or NAA) had a synergistic effect on shoot multiplication in *Sorbus commixta* Hedl. [58].

The juvenile phase of most plants inherently has a higher rooting potential than the mature phase [27]. Shoots from juvenile seedlings had a higher rooting ability than those from mature material in *Diospyros kaki* L. [63]. Physiological status also played a significant role in the rooting ability of *Sorbus* species [34,37,41,43].

In vitro culture can conduce rejuvenation (or reinvigoration), but not always, and it can depend on the number of subcultures in some species. In vitro serial subculture improved rooting in *Diospyros kaki* [63], *Eucalyptus urophylla* S. T. Blake [64], apple and cherry rootstocks [65].

Root formation in microcuttings depends on the plant species (genotype) and the cultivation method applied. Even within a species, the rooting ability of microcuttings varies among different cultivars and clones [66]. The effect of genotype was also observed in four *Sorbus* species; significant differences in rooting ability were found between *S. × abscondita*, *S. gemella*, *S. × kitaibeliana* and *S. omissa* (Figure 2A). Regardless of the age of the donor plants, *Sorbus* × *abscondita* rooted best even when a donor tree was about 50 years old, while *S. gemella* had the lowest rooting ability of all *Sorbus* species even when the in vitro culture was established from a 10–15-year-old tree. The opposite situation was found in *Sorbus* × *kitaibeliana*, where rooting ability was low (an approximately 80-year-old tree), while *S. omissa* had a high rooting ability (a 10-year-old tree). These results suggest that genotype plays a large role in rooting ability. *Sorbus* × *abscondita* is a member of *S.* subgen. *Soraria* Májovský and Bernátová [67] originating from the hybridization of some members of *S.* subgen. *Aria* and *S. aucuparia*. In contrast, *S. gemella*, *S*. × *kitaibeliana* and *S. omissa* come from the *S.* subgen. *Tormaria* Májovský and Bernátová [67], where instead of *S*. *aucuparia*, the second parent is *S. torminalis*. Since the two parent species are quite different ecologically and genetically, the newly created species can show significantly different characteristics accordingly.

One positive aspect of in vitro cultures is that woody species that do not root easily via conventional propagation methods can easily be rooted as microcuttings [68,69,70]. In general, the problems of rooting are more pronounced with woody species [71]. Physiological status played a significant role in the rooting ability of some *Sorbus* species. Microcuttings derived from juvenile material had a higher rooting ability than those from mature material [34,37,41].

The rooting of microcuttings can take place either in vitro or ex vitro. Rooting in ex vitro environments has two main benefits: it is economic and functional. Rooting under non-sterile conditions should reduce the cost of plant production by microcutting the steps of aseptic manipulation [44]. In vitro rooting of micropropagated plants is an expensive process that can double the final price of these plants [72].

The formation of roots in a tissue culture environment, especially under high humidity and in the presence of sugar, results in morphological and physiological differences compared to roots that develop under normal conditions [73,74,75]. Therefore, in vitro roots must adapt in non-sterile conditions to function properly. By contrast, ex vitro rooted plantlets did not require any additional acclimatization prior to transplanting to regular greenhouse conditions [76].

Ex vitro rooting includes two basic methods to root microcuttings. Both the induction and expression of roots are performed either ex vitro in a greenhouse medium, or they take place in agar or a liquid culture within an ex vitro environment [27].

In vitro rooting of microcuttings has been reported for many *Sorbus* species and their infrageneric hybrids: *S. domestica* [34,37], *S. aucuparia* [43], *S. torminalis* [30], *S. commixta* [58], × *Malosorbus florentina* (Zuccagni) Browicz [41] and also seven rare and endemic *Sorbus* species [42]. By contrast, ex vitro rooting has been reported only from an experiment with *S. torminalis*, where root induction took place in vitro and expression occurred under greenhouse conditions [77].

In previous experiments, we tested the rooting potential of *Sorbus × abscondita* and *S. omissa* on a half-strength MS agar medium containing IBA or NAA (α-naphthalene acetic acid) at concentrations of 0, 0.5 and 3.0 mg·L^–1^. The highest rooting (55.6%) was observed in *S. × abscondita* at 3 mg·L^−1^ of IBA. In *S*. *omissa*, no roots were formed, regardless of the type of auxin and its concentration [78]. When these genotypes were rooted ex vitro, there was a significant increase in rooting ability in *S. × abscondita* (74%) and in *S. omissa* (62%—Figure 2A). Similar results were observed in mulberry [68].

Auxin is a key growth regulator in adventitious root formation in plants. Many plants require the presence of auxin for efficient root regeneration [79]. Some of the most important factors affecting in vitro adventitious rooting are the choice of auxin, its concentration and the duration of tissue exposure [80]. IBA is the most often-used agent for rooting microcuttings of a wide variety of species [27]. When microcuttings are rooted in vitro, not only does the level and the duration of auxin treatment differ from ex vitro rooting, but so does the rate of gas exchange by the section of the stem in which the new roots are formed. The presence of auxin in the rooting medium increases the synthesis of ethylene and can have a negative effect on root regeneration [81,82].

The use of various auxins for in vitro root formation in *Sorbus* species has been reported as follows: Rhizogenesis was induced with IBA in *S. commixta* and in *S.* × *abscondita* [58,79]; in *S. domestica*, a high percentage of rooting was achieved with both IBA and NAA. A combination of IBA with NAA proved effective in *S. torminalis* and *S. aucuparia* [32,43], and a combination of IBA with IAA was found to have an effect in infrageneric hybrid × *Malosorbus florentina* [41] and NAA has been used with success in seven rare endemic species [42].

In our experiments, we tested the possibility of ex vitro rooting (performing both induction and expression) in four *Sorbus* species. Rooting was achieved in all the species tested. Rooting ability was high in *S. × abscondita* and *S. omissa* but lower in *S. gemella* and *S.* × *kitaibeliana* (Figure 2A). Regarding the effect of auxin treatment on rooting ability, very interesting results were achieved: in *S. gemella*, *S.* × *kitaibeliana* and *S. omissa*, better results were obtained in rooting without auxin treatment, excluding *S.* × *abscondita*, where the highest rooting percentage was observed in the 2% IBA treatment (Figure 2G). A reduction in rooting in the presence of auxin in an agar medium compared to the control (no auxin) has been reported for *Anemone* L. [83]. The balance of endogenous growth hormones is a key factor in successful organogenesis in plant tissue cultures. Hormonal levels are significantly affected by the addition of exogenous PGRs to the culture medium [84]. The application of PGRs during multiplication can have a negative effect on subsequent rhizogenesis, which has been demonstrated, for example, in *S. torminalis* [30].

Another factor that can influence rooting in woody plants is the size of the cutting/microcutting [85,86]. Our results showed that determining the optimal length of microcuttings in *Sorbus* species had a positive effect on rooting. In general, shorter shoots were thinner than longer shoots. The leaves were also less mature. In *S*. × *abscondita*, the biggest differences were observed visually in the maturity of the leaves; in addition, shorter shoots were more sensitive to dying.

In conclusion, a successful micropropagation protocol for four *Sorbus* species was developed. The most difficult phase of the protocol was root induction. The results showed that *Sorbus* species can be rooted non-sterile in a peat substrate without the presence of auxin. Compared to the other species, *Sorbus* × *abscondita* had the highest regeneration and rooting ability. The significantly better results we achieved may have been caused by the influence of a different parental combination of the mentioned species. This protocol will be used in future studies to evaluate the regeneration potential in other species of *Sorbus* subgen. *Tormaria* and *Sorbus* subgen. *Soraria* [67].

## 4. Materials and Methods

### 4.1. Plant Material and In Vitro Culture Initiation

Characteristics of *Sorbus* species used in our study (Figure 5) and their localities are as follows:

*Sorbus* × *abscondita* is a rare hybrid between *S. aucuparia* and *S. danubialis* [14] that is treated as a member of *S.* subgen. *Soraria* [67]. To this day, it has been recorded at eleven localities in Bohemia (historical part of Czechia) [14], but five of them have gone extinct. Except for two populations, it grows only individually at each locality. The coordinates of the localities of living plants of this taxon are as follows: population in Kladno-Švermov town (Kladno district)—50°10′13.0″ N, 14°07′32.0″ E; population on Sedlo hill near the town of Úštěk (Litoměřice distict)—50°35′31.0″ N, 14°15′43.0″ E; Pochvalovská stráň slope near the village of Pochvalov (Rakovník district)—50°14′11.0″ N, 13°49′03.1″ E; Solopysky village (Louny district)—50°16′08.5″ N, 13°44′25.7″ E; Stříbrník hill near the village of Úholičky (Praha-západ district)—50°10′02.6″ N, 14°21′03.4″ E; Trmice town (Ústí and Labem district)—50°38′07.8″ N, 14°00′44.3″ E.

*Sorbus gemella* is a triploid species [87] regarded as a member of *S*. subgen. *Tormaria* [67], with the supposed parentage of *S. danubialis* and *S. torminalis*. Its distribution is limited to the Džbán tableland hills located mostly in the Louny district in the north-central part of Bohemia [14]. The species inhabits edges of marlite plateaus and upper parts of steep slopes on the same bedrock predominantly in oak woodlands. Several subpopulations consist of approximately 1500 individuals. The approximate GPS coordinates of the type locality in Konětopy are 50°16′5.4″ N, 13°44′22.3″ E.

*Sorbus* × *kitaibeliana* is a hybrid with the supposed parentage of *S. danubialis* and *S. torminalis* (a member of *S.* subgen. *Tormaria*). Its only known locality in Czechia is in northern Bohemia near the town of Trmice (Ústí and Labem district—50°38′25″ N, 14°01′09″ E) [14]. There is one known adult, a 12 m high tree, and one 1.5 m high seedling.

*Sorbus omissa* is another triploid species [88] of the same subgenus and supposed parentage. It is a stenoendemic species occurring in central Bohemia, in the vicinity of the towns of Roztoky and Libčice nad Vltavou (Praha-západ district) in the valley of the lower reach of the river Vltava, where it grows primarily in oak woodlands on humus soils on a substratum of Proterozoic slate. Its two known populations consist of approximately 150 individuals [14,88]. The GPS coordinates of the type specimen of the species are 50°10′06.9″ N, 14°21′09.5″ E.

**Figure 5 plants-12-00488-f005:**
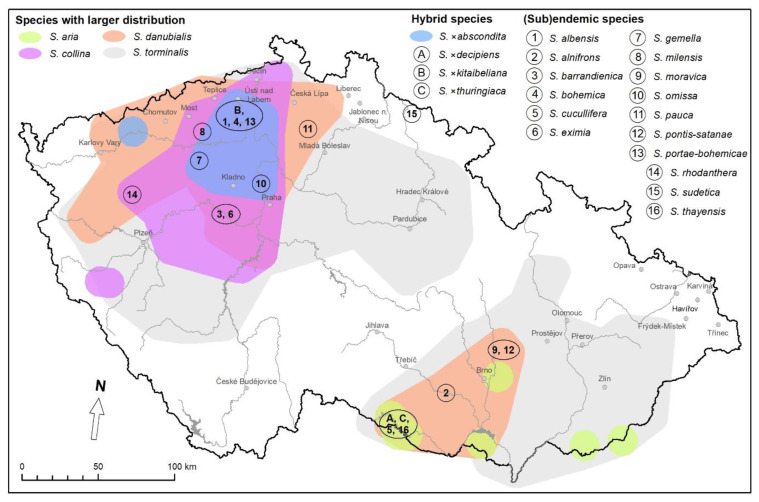
Map showing a distribution of *Sorbus* taxa in Czechia except for *Sorbus aucuparia* (it is widespread throughout the country). The areas of species with larger distribution were delineated with the “Sample by Buffered Local Adaptive Convex-Hull” tool [89], which combines the creation of a wrapper zone and a minimal convex polygon. Data on *Sorbus* species with larger distribution were derived from the Pladias database [14,90].

All tested plants were triploids (2n = 3x = 51). The DNA ploidy level was estimated using flow cytometry [5]. *Carex acutiformis* Ehrh. was used as an internal standard.

Plant material of the four *Sorbus* species, with one clone from each of the selected samples (*S.* × *abscondita*—Kladno-Švermov, an approximately 50-year-old donor tree; *S. gemella*—Konětopy, an approximately 10–15-year-old tree; *S.* × *kitaibeliana*—Trmice, an approximately 80-year-old tree and *S. omissa*—Roztoky, an approximately 10-year-old tree; Figure 6), was identified and collected from natural localities in April or May. Shoots were taken from the lower part of the crown; epicormic sprouts were not available. To obtain in vitro cultures, stem cuttings were taken (40–60 mm in length, Figure 7A,B) and surface-sterilized in 2.5% sodium hypochlorite solution (50% Savo^®^, Bochemie a.s., Bohumín, The Czech Republic) for 15 min with one drop of Tween-20. Finally, the segments were washed with sterile distilled water three times for 10 min each. Cultures were initiated from apical stem segments with one apical bud and one pair of axillary buds (10–15 mm in length, Figure 7C) and transferred onto a full-strength solid Murashige and Skoog (MS) basal medium with vitamins [91] (Duchefa Biochemie B.V., RV Haarlem, The Netherlands) supplemented with 0.5 mg·L^−1^ BA (*N*^6^-benzyladenine), 0.1 mg·L^−1^ IBA (indole-3-butyric acid), 20 g·L^−1^ sucrose, 2 mL·L^−1^ PPM (Plant Preservative Mixture, Washington, DC, USA) and 7 g·L^−1^ agar (Sigma-Aldrich Inc., St. Louis, MO, USA). The pH was adjusted to 5.7 using NaOH prior to autoclaving. Explants were cultured in 100 mL Erlenmeyer flasks containing 25 mL of a MS medium. The cultures were maintained in a growth room under a 16 h photoperiod with a photosynthetic photon flux density (PPFD) of 60 µmol·m^−2^·s provided by cool-white fluorescent tubes (Tungsram, General Electric Company, Boston, MA, USA) at 22 ± 1 °C. Newly grown shoots were divided into two or three parts by subculturing to a fresh medium every three to four weeks. Multiplication and rooting experiments were performed on in vitro cultures that were at least one year old (Figure 8).

### 4.2. In Vitro Shoot Proliferation

We evaluated the effect of two cytokinins (BA, *m*T) alone at a concentration of 0.5 mg·L^−1^ and in combination with an auxin (IBA) at a concentration of 0.1 mg·L^−1^, and of IBA alone at the same concentration on the induction of shoot formation in four *Sorbus* species. The plant growth regulator-free medium (PGR) was used as a control. Single shoots (2–3 expanded leaves, ≥1.5 cm long) were excised from the stock cultures and multiplied on an MS-medium containing vitamins, 20 g·L^−1^ sucrose and 7 g·L^−1^ agar. Each treatment was repeated three times, with twelve explants per treatment. The cultivation conditions for this experiment were the same as above. The mean number of shoots per explant and total shoot length were recorded after five weeks.

### 4.3. Ex Vitro Rooting and Acclimatization

Shoots of the four *Sorbus* species were harvested from an 8–10-week-old in vitro culture maintained on a basal MS medium with 0.5 mg·L^−1^ BA and 0.1 mg·L^−1^ IBA, but without PPM. For rooting experiments, microcuttings of two different shoot lengths, 15–25 mm or 26–35 mm, were used. The microcuttings were treated with rooting powder (Rhizopon^®^AA, Rhizopon BV, Rijndijk, The Netherlands) at IBA concentrations of 1% or 2%.

The microcuttings were cut at the base with a sharp scalpel and immersed in 0.15% antifungal Previcur Energy (Bayer S.A.S., Lyon, France) for 1 min before the application of the rooting powder product (excluding the control). They were then inserted into a plastic dish (14 × 8.5 × 5 cm) with four holes for drainage of excess water, containing a steamed peat-perlite substrate (1:1, *v*/*v*) and watered with tap water. Each dish contained eighteen microcuttings. Three dishes were placed in a Minipa plastic box covered with a clear plastic cover with ventilation (tall lid model, Fima, Brno, The Czech Republic) and transferred to a growth room lit with cool-white fluorescent tubes (Tungsram, General Electric Company, Boston, MA, USA) and a photosynthetic photon flux density of 60 µmol·m^−2^·s^−1^ for a 16 h photoperiod at 24/19 ± 1 °C (day/night). Each treatment was repeated at least three times, with eighteen microcuttings, i.e., a total of 54 microcuttings per treatment. Root induction (%) and the mean number of roots per microcutting were recorded after six weeks. Then, the rooted plants were replanted into pots (Teku^®^ 10 cm diameter) with a peat substrate Remix-D (Rékyva, Siauliai, Lithuania) and transferred to a greenhouse at 23 °C under natural photoperiod conditions.

### 4.4. Statistical Analysis

Two-factor analysis of variance and Tukey’s multiple comparison method were used to evaluate differences between the explants. The factors in this case were the *Sorbus* species and the type of treatment. The dependent variables were the mean number of shoots per explant and the total length of all shoots in the explant. The mean number of shoots per explant could be considered a variable with a Poisson distribution, so the following transformation:(1)$$z=\sqrt{x+\frac{3}{8}}$$
was applied to it [92]. The total length of all explant shoots could be considered a variable with a normal distribution with sufficient accuracy, so its transformation was not necessary.

A three-factor analysis of variance (with the factors *Sorbus* species, treatment variant and microcutting size) followed by Tukey’s multiple comparison method was used to assess differences between microcuttings. For some dependent variables, the assumption of normal distribution was violated, and various transformations were applied. Specifically, the proportion of rooted microcuttings was a variable with a binomial distribution, so the following transformation:(2)$$z=\arcsin\sqrt{x}$$
was applied to it [92]. Again, the mean number of shoots per microcutting could be considered a variable with Poisson distribution, so it was transformed using (1). The results are presented using homogeneous groups at the 0.05 level of significance.

## Figures and Tables

**Figure 1 plants-12-00488-f001:**
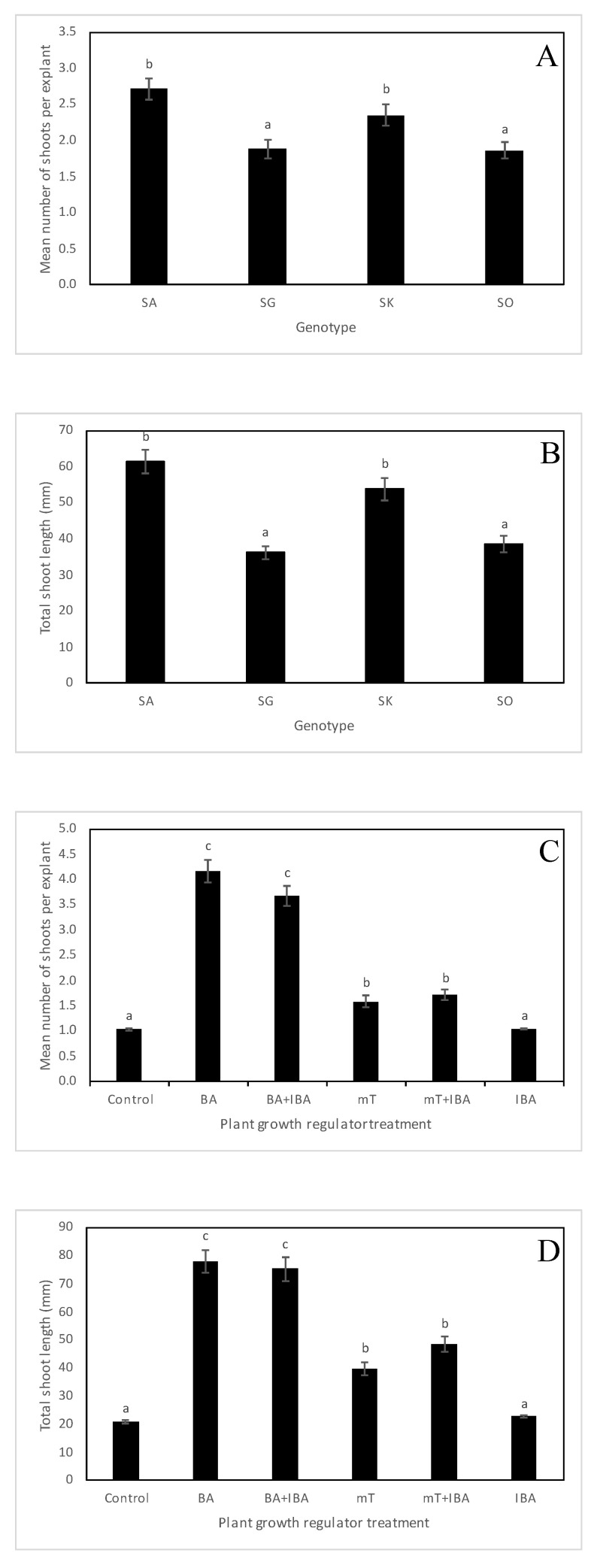
Shoot production in *Sorbus* species cultured with different combinations of *N*^6^-benzyladenine (BA), indole-3-butyric acid (IBA) and *meta*-topolin (*m*T), applied at 0.5 mg·L^−1^, 0.1 mg·L^−1^ and 0.5 mg·L^−1^, respectively. (**A**,**B**) The effect of genotype on the mean number of shoots per explant (**A**) and the total shoot length (**B**). (**C**,**D**) The effect of plant growth regulator (PGR) treatment on the mean number of shoots per explant (**C**) and on the total shoot length (**D**). Values are mean standard errors. In each graph, different letter(s) on the bars show significant differences according to Tukey’s range test (*p*-value = 0.05). *Sorbus × abscondita* (SA), *S. gemella* (SG), *S. × kitaibeliana* (SK), *S. omissa* (SO).

**Figure 2 plants-12-00488-f002:**
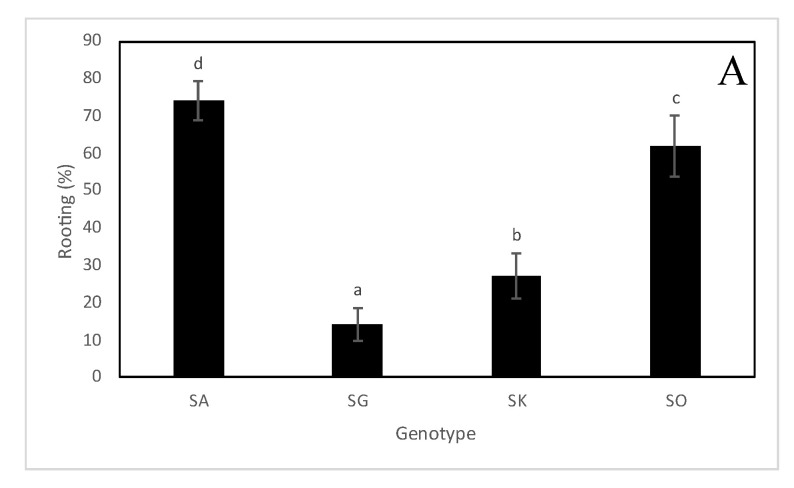
Rooting of microcuttings of *Sorbus* species treated with different concentrations of indole-3-butyric acid (IBA), applied at 0%, 1% and 2%. (**A**,**B**) The effect of genotype on the rooting percentage (**A**) and the mean number of roots per microcutting (**B**). (**C**,**D**) The effect of auxin treatment on the rooting percentage (**C**) and the mean number of roots per microcutting (**D**). (**E**,**F**) The effect of the size of the microcutting on the rooting percentage (**E**) and the mean number of roots per microcutting (**F**). (**G**,**H**) The interaction between the genotype and auxin treatment on the rooting percentage (**G**) and the mean number of roots per microcutting (**H**). (**I**) The interaction between the genotype and the size of the microcutting on the rooting percentage. Values are mean standard errors. In each graph, different letter(s) on the bars show significant differences according to Tukey’s range test (*p*-value < 0.05). *Sorbus × abscondita* (SA), *S. gemella* (SG), *S. × kitaibeliana* (SK), *S. omissa* (SO).

**Figure 3 plants-12-00488-f003:**
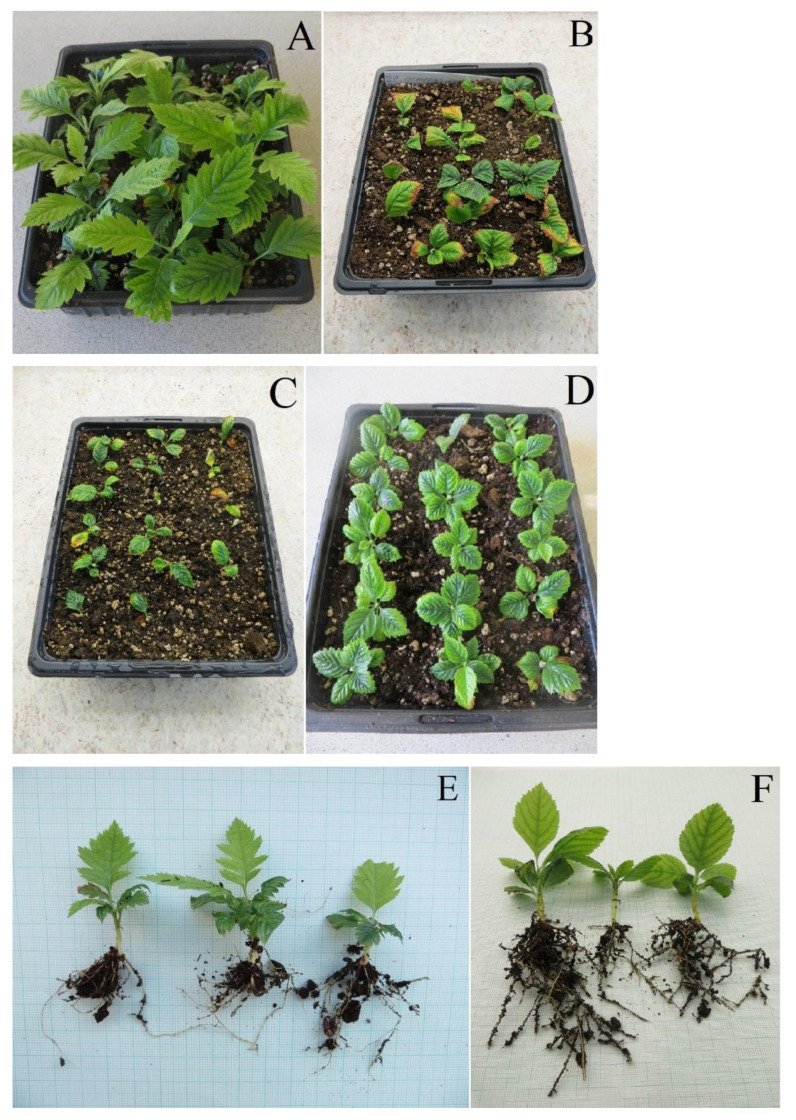
Comparison of the development of microcuttings in *Sorbus* species in the best rooting treatment in a steamed peat-perlite substrate after six weeks. (**A**) *Sorbus* × *abscondita*, microcuttings treated with 2% indole-3-butyric acid (IBA) rooting powder. (**B**) *Sorbus gemella*, the control. (**C**) *Sorbus* × *kitaibeliana*, the control. (**D**) *Sorbus omissa*, the control. (**E**) Detail of rooted plantlets of *Sorbus* × *abscondita*. (**F**) Detail of rooted plantlets of *Sorbus omissa*.

**Figure 4 plants-12-00488-f004:**
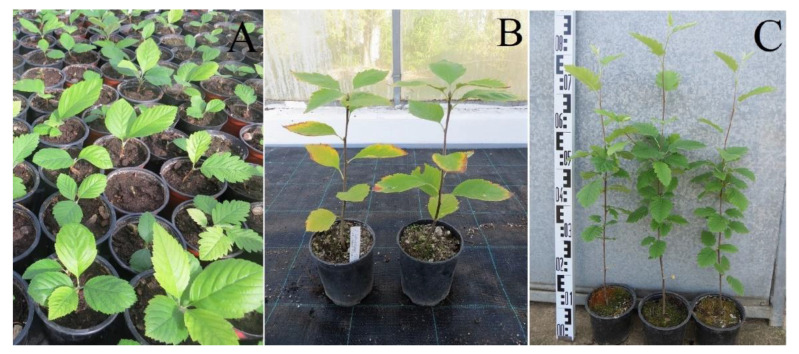
Acclimatization and development of rooted plants of *Sorbus* species. (**A**) 1-year-old plants (*S*. × *abscondita* and *S. omissa*). (**B**) 2-year-old plants (*S*. × *kitaibeliana*). (**C**) 3-year-old plants (*S*. × *abscondita*).

**Figure 6 plants-12-00488-f006:**
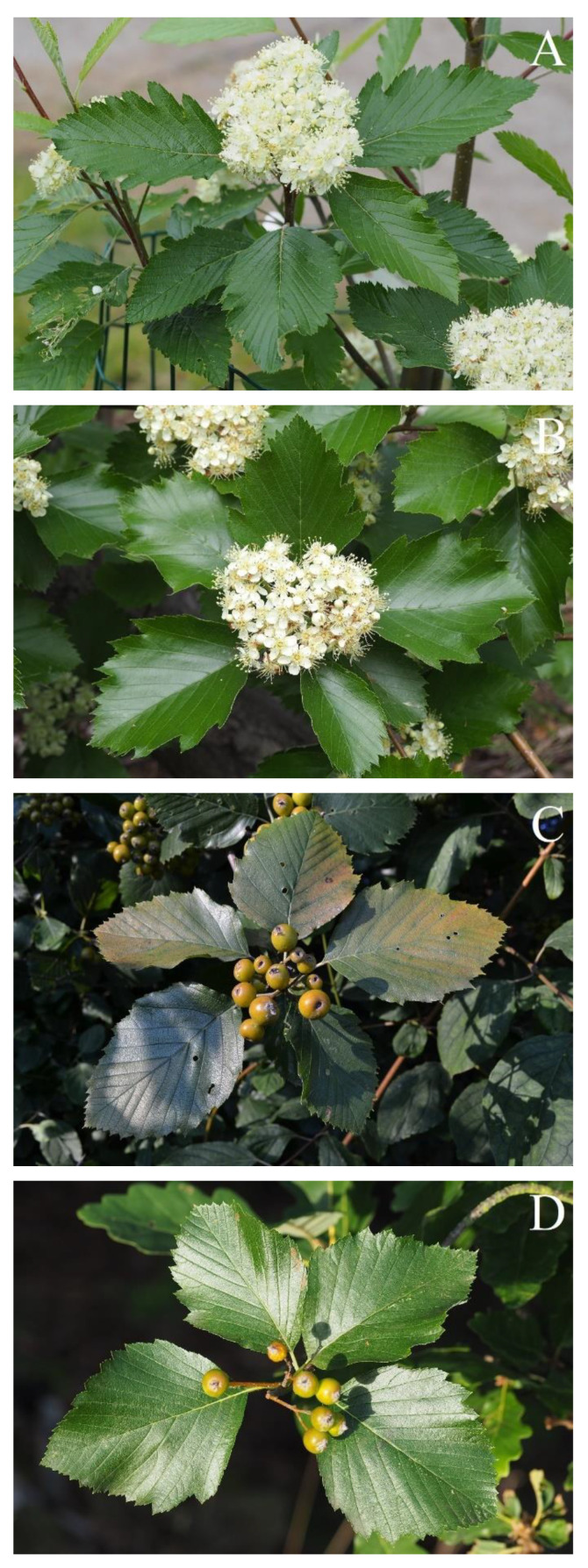
*Sorbus* species investigated. (**A**) *Sorbus* × *abscondita*. (**B**) *Sorbus gemella*. (**C**) *Sorbus* × *kitaibeliana*. (**D**) *Sorbus omissa*.

**Figure 7 plants-12-00488-f007:**
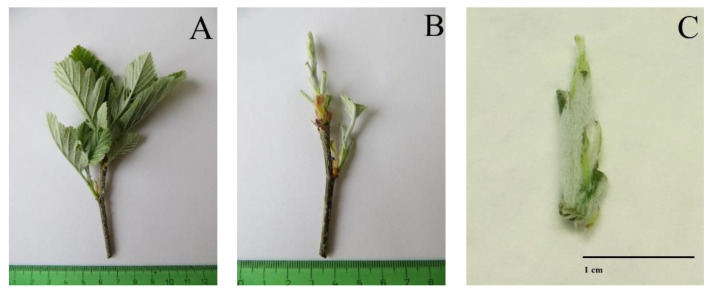
Explant source in *Sorbus* sp. (**A**) Original shoot. (**B**) Prepared shoot before explant disinfestation. (**C**) Initiation explant.

**Figure 8 plants-12-00488-f008:**
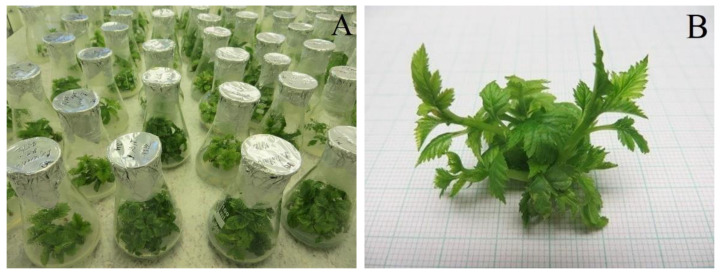
Bulking of stock material in *Sorbus* species on full-strength solid Murashige and Skoog (MS) basal medium with vitamins supplemented with 0.5 mg·L^–1^ *N*^6^-benzyladenine (BA) and 0.1 mg·L^–1^ indole-3-butyric acid (IBA). (**A**) In vitro collection of *Sorbus* species. (**B**) Detail of shoot regeneration in *Sorbus* × *abscondita*.

**Table 1 plants-12-00488-t001:** The effect of interaction between the genotype and the PGR treatment on the mean number of shoots per explant in *Sorbus* species (n). The term SE denotes corresponding standard error; group shows significant differences according to Tukey’s range test (*p*-value = 0.05).

PGR	Genotype											
	*S. × abscondita*			*S. gemella*			*S. × kitaibeliana*			*S. omissa*		
	n	SE	Group	n	SE	Group	n	SE	Group	n	SE	Group
Control	2.00	0.28	abcd	1.11	0.09	abcd	1.28	0.09	ab	1.00	0.00	a
BA	2.97	0.41	de	2.36	0.39	bcde	3.42	0.59	de	2.42	0.36	abcde
BA+IBA	2.42	0.33	abcde	1.69	0.22	abcd	2.42	0.33	abcde	2.11	0.27	abcd
*m*T	3.72	0.36	e	2.39	0.42	abcde	2.83	0.36	cde	1.89	0.14	abcd
*m*T+IBA	3.47	0.31	e	2.69	0.43	bcde	2.81	0.34	cde	2.13	0.37	abcd
IBA	1.69	0.33	abcd	1.03	0.03	a	1.36	0.12	abc	1.03	0.03	a

Plant growth regulator (PGR), *N*^6^-benzyladenine (BA), indole-3-butyric acid (IBA) and *meta*-topolin (*m*T).

**Table 2 plants-12-00488-t002:** The effect of interaction between the genotype and the PGR treatment on the total shoot length in *Sorbus* species (l). The term SE denotes corresponding standard error; group shows significant differences according to Tukey’s range test (*p*-value = 0.05).

PGR	Genotype											
	*S. × abscondita*			*S. gemella*			*S. × kitaibeliana*			*S. omissa*		
	l	SE	Group	l	SE	Group	l	SE	Group	l	SE	Group
Control	56.69	8.9	bcdef	22.97	1.0	a	31.69	3.6	abc	20.67	0.6	a
BA	59.92	8.3	cdef	40.56	5.3	abcde	71.00	12.9	ef	42.36	5.7	abcde
BA+IBA	54.94	7.8	abcde	30.50	3.2	abc	56.03	7.6	bcde	42.03	5.2	abcde
*m*T	68.13	6.5	def	50.53	5.5	abcde	61.81	5.9	cdef	36.08	1.9	abcd
*m*T+IBA	89.22	8.1	f	44.86	4.1	abcde	65.50	6.3	def	50.49	8.8	abcde
IBA	39.99	7.3	abcde	27.00	1.2	ab	36.31	3.9	abcd	24.25	0.8	ab

Plant growth regulator (PGR), *N*^6^-benzyladenine (BA), indole-3-butyric acid (IBA) and *meta*-topolin (*m*T).

**Table 3 plants-12-00488-t003:** Effect of different factors (genotype, auxin treatment and microcutting size) on rooting of *Sorbus* species used in our study.

	Rooting				Mean Number of Roots per Microcutting			
	df	Mean Squares	F		df	Mean Squares	F	
Genotype	3	2.42	53.98	***	3	23.06	72.66	***
Auxin treatment	2	1.13	25.31	***	2	5.83	18.37	***
Microcutting size	1	1.15	25.71	***	1	1.52	4.80	*
Genotype × auxin treatment	6	0.35	7.97	***	6	3.98	12.56	***
Genotype × microcutting size	3	0.25	5.65	**	3	0.48	15.14	ns
Auxin treatment × microcutting size	2	0.02	0.59	ns	2	0.10	0.32	ns
Error	54	0.04			568	0.31		

*, **, ***, ns—Tukey’s range test significant at *p* = 0.05, *p* = 0.01, *p* = 0.001 or not significant; df means degrees of freedom, and F is the F-statistic.

## Data Availability

The data used in this work are new and original and are fully reported in the present manuscript.

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
