# Peer review of "Micropropagation as a Tool for the Conservation of Autochthonous Sorbus Species of Czechia"

_plants, 2023, doi:10.3390/plants12030488_

Round 1
Reviewer 1 Report
In this paper authors develope a protocol to micropropagate Sorbus species from Czechia, two of which are endemic. Authors ares sucessful in designing a protocol for each species. Although the work is interesting to preservate the species, it is more a technical paper and it does not have scientific discussion. Maybe it is more suitable for other type of journal with a more technical background. Regarding methodology aspects, it has no sense for me to compare the regeneration between species, because it does not matter if they perform differently, authors just have to decide the best methodology for each species.
Author Response
Thank you very much for your time and effort you’ve spent to improve my manuscript. I incorporated all suggestions and recommendations. Co-authors participated in the revision of the manuscript. Changes in the manuscript were marked using the “Track Changes” function. The manuscript has undergone English language editing by MDPI.
On behalf of all authors, yours sincerely,
Jana Šedivá
Head of the research teams

Reviewer 2 Report
The manuscript involves a well described and visually presented protocol for micropropagation of endemic Sorbus tree species. The work has practical application, and from fundamental point of view it is a nice example for presentation of procedure for ex vitro preservation of plant species of interest. Although some additional assays on physiological parameters would be interesting to be tested, the work could be published in the present study as a nicely organized in vitro and ex vitro cultivation protocol. The data are important for conservation of endemic tree species, and they could be of interest for experts working with other tree species, and for general experts in micropropagation.
Additional remarks:
Lines 99-132 – most the information is rather suitable for the material section, specially the geographic coordinates. Probably put in 4.1. section instead of “(see chapter 1. Introduction for characteristics of species and localities)”.
Line 232, Table 1 – explain abbreviations “df” and “F”, e.g. in the table’s footnote.
Line 257, Figure 4 – make the figure more compact, e.g. on one row A-D, and on second row – E-F.
Line 266, Figure 5 – make the figure more compact, e.g. on one row A-C.
Author Response
Thank you very much for your time and effort you’ve spent to improve my manuscript. I incorporated all suggestions and recommendations. Co-authors participated in the revision of the manuscript. Changes in the manuscript were marked using the “Track Changes” function. The manuscript has undergone English language editing by MDPI.
Lines 99-132 – most the information is rather suitable for the material section, specially the geographic coordinates. Probably put in 4.1. section instead of “(see chapter 1. Introduction for characteristics of species and localities)”.
Information regarding individual Sorbus species has been moved to Section 4. Materials and Methods.
Line 232, Table 1 – explain abbreviations “df” and “F”, e.g. in the table’s footnote.
The abbreviations “df” and “F” were explained in the table's footnote (Table 1).
Line 257, Figure 4 – make the figure more compact, e.g. on one row A-D, and on second row – E-F.
Figure 4 has been condensed into rows: A–B, C–D and E–F.
Line 266, Figure 5 – make the figure more compact, e.g. on one row A-C.
Figure 5 has been condensed into row: A–C.
On behalf of all authors, yours sincerely,
Jana Šedivá
Head of the research teams
Reviewer 3 Report
The findings reported by the authors are of broad interest. The authors present original results on developing a micropropagation protocol for 4 Sorbus species, involving two endemic ones: S. gemella and S. omissa and two hybrids: S. x abscondita and S. x kitaibeliana. Successfully, rooting was achieved in all species tested, although the differences in rooting ability were described for particular species. The comparison of the effectiveness of two cytokinins, BA and mT showed that BA had a significantly higher effect on shoot regeneration.
The results could possibly be very useful for future maintenance in the natural environment. The manuscript presents the topic, which is adequately solved. This paper is organised correctly and written clearly using correct grammar. The problem is adequately stated and solved. The results are presented in a clear and transparent way. Interpretations of the results are sufficient. I recommend accepting the manuscript after minor revision.
I found a few substantive and editing weaknesses:
The Authors stated that all tested plants are triploids (line 112). How it was analysed previously? No methods and proper literature is included. It should be supplemented.
The characteristic of Sorbus species used in the study and their localization do not include references (Introduction).
The protocol for other Sorbus species is known already. The Authors should add information about it in the Introduction.
Why do authors show photos of all Sorbus species (Figure 6) at different stages of development?
Author Response
Thank you very much for your time and effort you’ve spent to improve my manuscript. I incorporated all suggestions and recommendations. Co-authors participated in the revision of the manuscript. Changes in the manuscript were marked using the “Track Changes” function. The manuscript has undergone English language editing by MDPI.
The Authors stated that all tested plants are triploids (line 112). How it was analysed previously? No methods and proper literature is included. It should be supplemented.
This information has been added: All tested plants were triploids (2n = 3x = 51). The DNA ploidy level was estimated using flow cytometry [5]. Carex acutiformis Ehrh. was used as an internal standard.
The characteristic of Sorbus species used in the study and their localization do not include references (Introduction).
This information has been added in chapter Introduction: The characteristic of Sorbus species used in the study and their localization are presented in chapter 4. Materials and Methods.
The protocol for other Sorbus species is known already. The Authors should add information about it in the Introduction.
This information has been added: Research carried out in the last decade have shown that micropropagation is an important method for the propagation of rare and endemic Sorbus species as ×Malosorbus florentina [41], and then S. alnifrons, S. bohemica, S. gemella Kovanda, S. hardeggensis, S. quernea, S. rhodanthera and S. sudetica [42].
Why do authors show photos of all Sorbus species (Figure 6) at different stages of development?
We did not have available photos of individual Sorbus species at different stages of plant development (1st-3rd year), so we combined the photos.
On behalf of all authors, yours sincerely,
Jana Šedivá
Head of the research teams
Round 2
Reviewer 1 Report
Dear authors
I think that the paper can be accepted for publication in the Journal after minor corrections done by the same
Author Response
We are very glad that you considered our study for publication after major revision in Plants. Thank you very much for your time and effort you’ve spent to improve my manuscript. I incorporated all suggestions and recommendations. Co-authors participated in the revision of the manuscript. Changes in the manuscript were marked using the “Track Changes” function. The manuscript has undergone English language editing by MDPI.